# Closed-loop optogenetic control of cell biology enables outcome-driven microscopy

Josiah B. Passmore [1,2,4], Alfredo Rates [1,4], Jakob Schröder [1], Menno T. P. van Laarhoven [3], Vincent J. W. Hellebrekers[1], Henrik G. van Hoef [1], Antonius J. M. Geurts[1], Wendy van Straaten[1,2], Wilco Nijenhuis[1,2], Florian Berger [1], Carlas S. Smith [3], Ihor Smal[1] & Lukas C. Kapitein [1,2] ✉

Smart microscopy is transforming biological imaging by integrating real-time analysis with adaptive acquisition to enhance imaging efficiency. Whereas many emerging implementations are event-driven and focus on on-demand data acquisition to reduce phototoxicity, we here present 'outcome-driven' microscopy, a framework combining smart microscopy with optogenetics to control cell biological processes and achieve predefined outcomes. We validate this approach using light-based control of cell migration and nucleocytoplasmic transport, demonstrating robust spatiotemporal control of cellular behaviour in single cells and in cell populations.

Biological imaging is constantly evolving, with "smart" microscopes paving the way for efficient, adaptive workflows. A smart microscopy platform performs real-time analysis, adjusting acquisition parameters on the fly[1–6]. Typical implementations of such real-time feedback loops use the information from the acquisition to change microscope behaviour, such as imaging modalities, speed, or dimensions. This allows the microscope to adapt to the sample and minimise imaging that does not contribute to answering the biological question, thus preserving sample health by reducing phototoxicity and improving efficiency[1,5,7,8]. As such, the majority of smart microscopy implementations record the sample passively, for example, to track objects[9–13], continually optimise imaging parameters[10,14–17] or, in the case of event-driven microscopy, adjust modalities in response to specific events[18–20].

Beyond observation of a biological process, microscopy can also facilitate active control. Most notably, optogenetics provides a platform for inducible control of biological systems across scales[21–25] using light, with a dose-response relationship[26–29] and a rapid localised response of cells to spatial light patterns[30–33]. Optogenetics has been previously combined with closed-loop smart microscopy platforms in a passive manner, with the controller adjusting hardware on the fly through automated segmentation or tracking[34–38]. However, the control of a biological system available through optogenetics provides the opportunity to extend feedback control to optimise not only hardware in passive observation, but also actively control biological processes

themselves to achieve user-defined outcomes[22,39]. Indeed, combinations of closed-loop control and optogenetics have previously enabled such "outcome-driven" research, adjusting light intensity or patterns on the fly in vivo[40–47], or in cultured cells[48–63] as demonstrated in cybergenetics[64]—most often with specific custom platforms designed for intensity optimisation at the population or whole-cell level for gene expression. As such, none of these examples fully harness the potential of optogenetics to achieve automated subcellular control of cellular dynamics. Such control over biology has been a long-term goal of cell biological research[65].

Here, we present a generalised platform for outcome-driven microscopy that uses optogenetics and real-time feedback to achieve automated spatiotemporal control of subcellular cell biology. This platform can control both the spatial patterning and intensity of light and adjusts them on the fly to bring biological systems to predefined outcomes, as demonstrated by long-term guidance of cell migration over predefined paths and by the controlled titration of protein levels in the cytosol or nucleus.

## Results and discussion

### A flexible smart microscopy platform for outcome-driven experiments

First, we built a modular smart microscopy platform, with interchangeable modules for microscope communication strategies, image

[1]Cell Biology, Neurobiology and Biophysics, Department of Biology, Faculty of Science, Utrecht University, Utrecht, The Netherlands. [2]Centre for Living Technologies, Alliance TU/e, WUR, UU, UMC Utrecht, Utrecht, The Netherlands. [3]Delft Center for Systems and Control, Delft University of Technology, Delft, The Netherlands. [4]These authors contributed equally: Josiah B. Passmore, Alfredo Rates. ✉e-mail: l.kapitein@uu.nl

processing pipelines, and control approaches (Supplementary Fig. 1 and Supplementary Note 1). This platform can be adapted for a variety of smart microscopy experiments beyond our implementation by adjusting the modules to fit specific cases. For our implementation of outcome-driven microscopy, a feedback loop is established whereby the microscope captures an image for processing, and based on the information extracted from the image, the control module optimises specific biological processes to achieve predefined outcomes by adjusting the illumination parameters of a spatial light modulator.

## Directing single-cell migration with real-time feedback and optogenetic control

To test the capabilities of outcome-driven microscopy, we used cell migration as our first proof-of-concept. Control of cell polarisation and migration using optogenetics is well established through the use of photoactivatable Rho GTPases or recruitment of upstream effectors to

the plasma membrane[66–72] (Fig. 1a). Light directed migration has been achieved by gradient illumination[70], manually updating the area of illumination (AOI)[67–69,73,74], or with smart microscopy, using live segmentation with tracking to achieve constant illumination in the same cellular area[34–36]. We reasoned that outcome-driven microscopy could push this further, and constantly update the AOI to direct cells to predefined migration paths. This approach would allow us to overcome stochastic variations such as differences in protein expression levels, morphology, or response times, with the controller ensuring that cells stay on the predefined track. To establish this, we used feedback on the position of the cell centroid relative to a path made up of setpoints, combined with a trajectory tracking controller to selectively illuminate the cell in the region closest to the next selected setpoint (Fig. 1b and Supplementary Note 1). For reliable segmentation, we used the pre-trained AI segmentation method Segment Anything Model (SAM)[75], with a custom extension of its interface for tracking

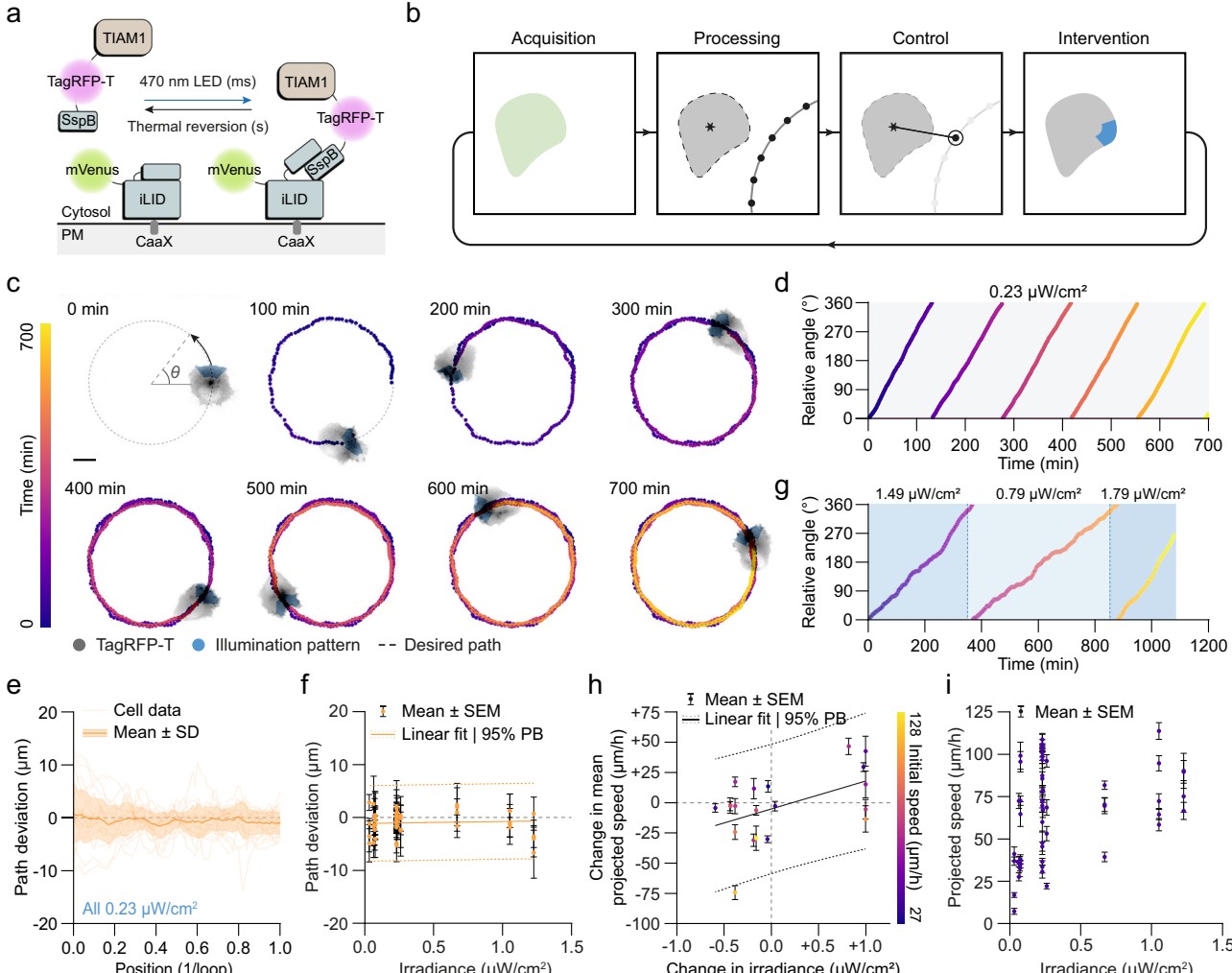

**Fig. 1 | Outcome-driven microscopy enables directed cell migration.**
**a** Schematic of the construct used for optogenetic recruitment of TIAM to the plasma membrane[72]. **b** Outcome-driven microscopy approach to control directed cell migration using a direct orientation-correction controller. **c** A HT1080-TIAM cell (grey) controlled using outcome-driven microscopy to illuminate a specific region of the cell (blue) and induce migration in (**a**) circle path (dashed line). The cell centroid tracked and marked (colour bar). Scale bar: 10 μm. **d** The angle of the cell centroid in (**c**) over time, relative to the starting position. **e** The path deviation of all cells guided to circle paths, with a constant irradiance of 0.23 μW/cm², separated by individual loops. n = 24 cells. Error bands show mean ± standard deviation (SD). **f** Average path deviation of all cells guided to circle paths with

various irradiance levels. n = 58 cells. Error bars show mean ± standard error (SEM). Error bands show linear fit with 95% prediction band (PB). **g** The angle of the cell centroid relative to the starting position for a cell with changing irradiance during the acquisition. **h** The change in mean projected speed for cells that were subjected to a change in irradiance between individual loops during guidance. Initial speed denotes the mean projected speed of the cell in the previous loop. n = 20 cells. Error bars show mean ± standard error (SEM). Error bands show linear fit with 95% prediction band (PB). **i** Average projected speed of each individual loop of controlled cells, per irradiance. n = 58 cells. Error bars show mean ± standard error (SEM). Source data are provided as a Source Data file.

(Supplementary Note 2). Using the migratory cell line HT1080 stably expressing an optogenetic construct to recruit the RAC1 effector TIAM1 to the plasma membrane[72], we found that we could reliably guide cells around a specific path multiple times for >10 h (Fig. 1c and Supplementary Movie 1), maintaining a relatively consistent speed (Fig. 1d and Supplementary Fig. 2a). Quantifying the distance between the centroid and the desired path (path deviation) revealed that the centroid of the cell was kept within 2.5 μm of the path (Supplementary Fig. 2b), demonstrating precise control of directed cell migration.

### Robust feedback control across heterogeneous cell populations

Outcome-driven microscopy provides the opportunity to control multiple cells in the same manner to the same setpoint, while also considering cell-cell variation. Indeed, when controlling multiple cells with the same constant LED irradiance (0.23 μW/cm²), we observed some variation in cell speed, independent of the expression level of TIAM1 (Supplementary Fig. 2c, d). Despite this heterogeneity, the controller was able to keep all cells within 5 μm of the desired path throughout the experiment (Fig. 1e). Path deviation was consistently low between cells, even at the lowest irradiance (Fig. 1f and Supplementary Fig. 2e), indicating that even a small degree of optogenetic activation is sufficient to maintain a consistent signalling pathway and guide cells to specific paths with outcome-driven microscopy.

### Adaptive modulation of migration speed via optogenetic feedback

Next, we asked if, despite cell-cell variation in speed when keeping irradiance constant, we could modulate cell speed by adjusting irradiance over time, due to the dose-response relationship of optogenetics. To evaluate this relationship, we guided cells around a circular path, and adjusted irradiance in a stepwise manner, changing the value for each complete or half loop (Fig. 1g and Supplementary Movie 2). Indeed, a greater variation in speed is seen when guiding cells with different light levels (Supplementary Fig. 2f), and notably, we observed some degree of irradiance-dependence of relative cell speed (Fig. 1g, h), demonstrating that adjusting irradiance can speed up or slow down cells. Interestingly, the inherent variability in the baseline speed of unstimulated, undirected cells (Supplementary Fig. 2g) prevented us from consistently achieving specific absolute speed setpoints at fixed irradiance levels (Fig. 1i). We observed that cells that are initially faster are more capable of decreasing their speed, and cells that initially move slow are easier to speed up, indicating a saturation point. Overall, the non-linear relationship we have observed between irradiance and control of speed may reflect underlying mechanisms or regulatory processes that buffer against external modulation. Variability in factors such as expression levels of regulators, polarity components, or metabolic state could influence how each cell interprets or responds to optogenetic input. Additionally, feedback within migration signalling networks may limit the extent to which light-induced perturbations can push cell behaviour beyond certain physiological constraints. As prolonged blue light exposure and sustained TIAM activation could also contribute to the heterogeneity that we observed by perturbing cell behaviour or inducing cell death, we evaluated phototoxicity by exposing HT1080-TIAM cells to whole-field illumination at low irradiance for 24 h. A minimal increase in toxicity was detected compared to non-exposed controls on the same coverslip, indicating that phototoxicity is not responsible for the observed heterogeneity and supporting the feasibility of long-term experiments (Supplementary Fig. 2h–j).

### Parallel control of multiple migrating cells with real-time coordination

To demonstrate the high level of control that we can achieve and to push the limits of outcome-driven control of cell migration, we adapted our platform for outcome-driven microscopy to control multiple cells simultaneously, each on a different path. Additionally, to avoid collisions we added coordination between cells by incorporating a lookahead, whereby the cells will be pulled back in the direction opposing their initial movement if an imminent collision is detected (Fig. 2a). We observed that we could effectively guide multiple cells to their respective paths with minimal error, and effectively stall them to avoid collisions (Fig. 2b and Supplementary Movie 3).

Furthermore, we considered that a future application of this approach for investigating cell dynamics is to explore cell-cell interactions during migration. To test this, we controlled cells to directed paths in a crowded environment, in the presence of cells that are not expressing the optogenetic construct and observed cell interactions during directed migration that influenced both the migration of uncontrolled cells and the morphology of the controlled cell (Fig. 2c and Supplementary Movie 4). MCF7, an epithelial cell type that displays collective cell migration[76] would represent an interesting model for studying these interactions. As a preliminary step, we demonstrated that our method can successfully direct migration in single MCF7 cells (Supplementary Movie 5), establishing a foundation for future work investigating collective behaviour.

### Dynamic feedback control of protein localisation using optogenetic export systems

In addition to outcome-driven microscopy aimed at controlling whole-cell dynamics with spatial illumination patterns, we next set out to establish a subcellular proof-of-concept and explore an intensity-driven relationship in depth to further demonstrate flexibility in control strategy. The blue light-induced nuclear export system (LEXY)[77] has a rapid light-response, and induces striking subcellular localisation differences with a dose-response relationship (Fig. 3a). We hypothesised that we could achieve full control of protein levels within the cytosol and nucleus using outcome-driven microscopy by adjusting illumination irradiance on the fly, placing irradiance under control of a proportional-integral-derivative (PID) controller[78] (Fig. 3b). The controller is optimised to minimise the error of the intensity in either the nucleus or cytosol in comparison to a setpoint. To reliably segment both the nucleus and the cytosol, we used our custom interface for the pre-trained SAM[75] and implemented a temporal filter to resolve timepoints with low contrast between the intensity of the nucleus and the cytosol (Supplementary Note 2). To illuminate the cells, we chose whole-cell illumination to ensure all optogenetic modules would be maintained in their active state in both the nucleus and cytosol during irradiance.

### Model-informed controller design and gain scheduling

First, we measured cell-cell variation in nucleocytoplasmic transport rates to inform our control strategy. In U2OS cells expressing LEXY under a doxycycline-inducible promotor, we observed that the expression level was a major contributor to cell-cell variation, with higher expressing cells having a lower rate of export (Supplementary Fig. 3a). Interestingly, we find that import rates do not correlate with the expression level, as well as the steady-state ratio of intensities between the two compartments (Supplementary Fig. 3b, c). As extended control requires prolonged exposure to low levels of blue light, we also validated that these cells showed only a minimal increase in cell death compared to non-illuminated controls for long-term experiments (Supplementary Fig. 3d–f). After tuning a basic PID controller based on a model of this system (Supplementary Note 3), we were able to control cytosolic intensity to desired setpoints, and the error between measured intensity and setpoint was consistently <10% (Supplementary Fig. 4a–c). However, we found that due to our design constraints aimed at minimising overshoot, these model-derived gains were not universal enough to deal with the nonlinear dynamics and cell-cell variation, especially when dynamically changing setpoints across the whole range of cell intensities (the operating range). In these

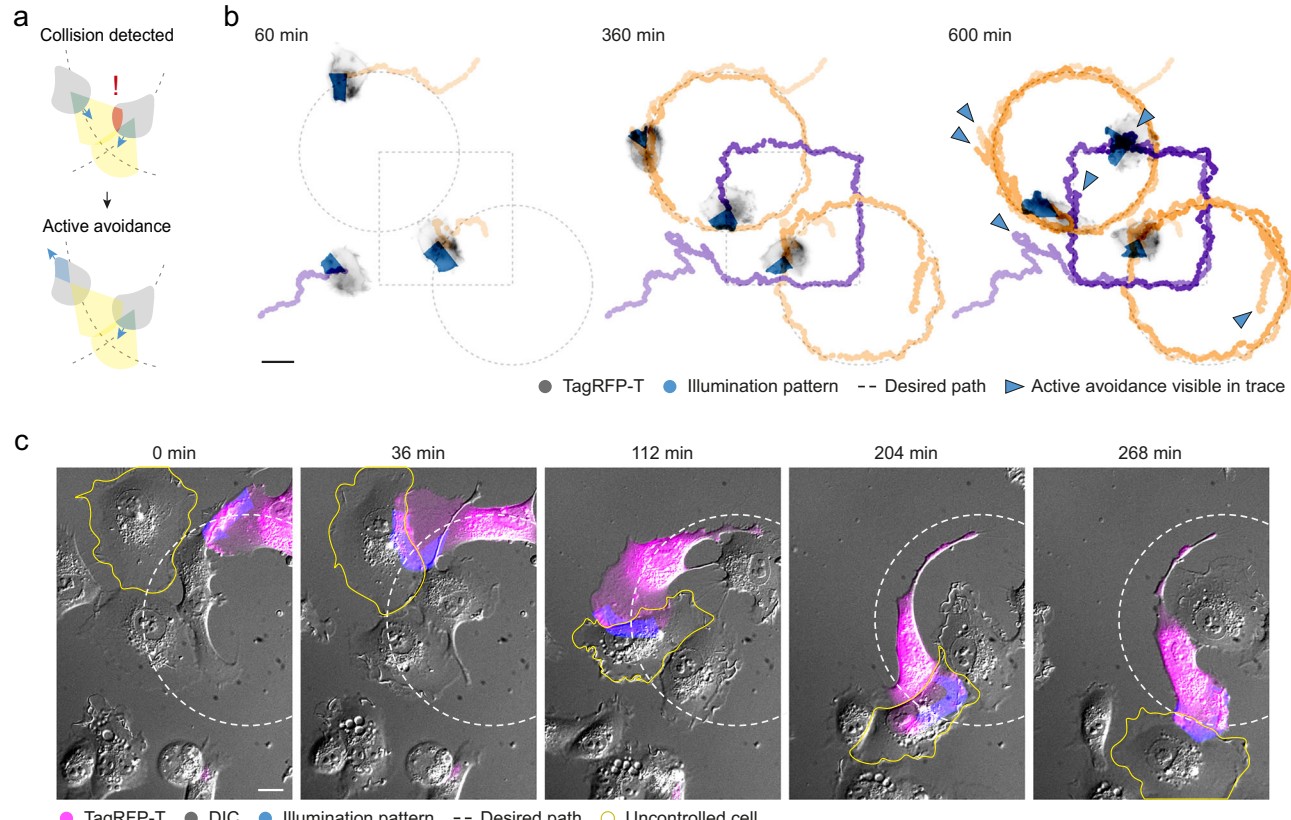

**Fig. 2 | Outcome-driven control of multiple cell migration and cell-cell interaction. a** Schematic of the active avoidance system for multi-cell outcome-driven control of directed cell migration. **b** Three HT1080-TIAM cells were simultaneously controlled to individual overlapping paths, with an active avoidance system as in (**a**) pulling back cells to ensuring collisions were avoided. Cell centroids were tracked and overlaid (orange and purple), with blue arrowheads showing points in the track where successful avoidance is visible. Scale bar: 20 μm. **c** A HT1080-TIAM cell controlled to a circle path in a crowd of WT HT1080 cells. The migration of an uncontrolled cell (yellow outline) is influenced by the movement and contact of the controlled HT1080-TIAM cell, and cell-cell contact drastically alters the morphology of the HT1080-TIAM cell. Scale bar: 10 μm.

cases, we often observed suboptimal behaviour such as steady-state offsets, depending on the cell (Supplementary Fig. 4d).

Therefore, we improved out controller by including gain scheduling, which adjusts PID gains as a function of a scheduling variable[79]—in our case, measured intensity—allowing gains to be optimised across the whole operating range. We implemented a gain-scheduled PID controller by first selecting a set of gains for various operating points, using our first-principle model of the system (Supplementary Note 3). We then fine-tuned the controller to find an optimal value-gain matrix. Using this gain-scheduled controller, we were able to control cells for various setpoints within the whole operating range to a high degree of accuracy, overcoming cell-cell variation and stochasticity, and outperforming the standard PID controller across the entire range (Fig. 3c and Supplementary Fig. 4e, f).

### Independent control of nuclear and cytosolic protein levels

To demonstrate flexibility in control strategies, and because we envision applications of outcome-driven control of nucleocytoplasmic transport that would benefit from control of nuclear intensity, we also developed a controller to modulate intensity levels in the nucleus (Supplementary Note 3). To compare the two controllers and to show robustness of control, we ran both the cytosol and nucleus controllers in the same cell, repeating three times for each controller (Fig. 3d and Supplementary Movie 6). Both controllers were highly reproducible, and were able to bring intensities to the predefined setpoints with remarkably similar irradiance levels, and with minimal error (Supplementary Fig. 5a–c). Interestingly, we see that the nucleus controller is less noisy than the cytosol controller, likely because the average

nucleus intensity is more representative of the real concentration than the average cytosol intensity is for the cytosol, due to the dynamic three-dimensional shape of the cell, the distribution of the marker across the cytosol, and changes of the segmentation through time. We were able to demonstrate long-term repeated control for many setpoints using the nucleus controller, without deviation in error over time (Fig. 3e–g and Supplementary Movie 7). The controller was able to maintain setpoints over extended periods, both when expression was increasing after addition of doxycycline, and after washing out doxycycline, so that expression is more stable (Supplementary Fig. 5d). We were also able to control COS-7 cells with the same implementation, to demonstrate flexibility in experimental design and broader applicability (Supplementary Fig. 5e).

### Simultaneous feedback control of protein localisation in multiple cells

Finally, we aimed to simultaneously modulate multiple cells to the same setpoint, showing further controller robustness and overcoming cell-cell variation. To achieve this, we established an alternative spatial light modulator setup that could control multiple cells simultaneously with varying intensities, illuminating individual nuclei with distinct irradiances over time using pulse-width modulation. To avoid unintended activation of cells in close proximity due to light scattering, we chose to illuminate a region in the centre of the cell for these experiments (Supplementary Fig. 6a, b). With this method, we could bring a group of seven cells to the same intensity value (Fig. 4a, b and Supplementary Movie 8). Notably, we could demonstrate that the controller maintains distinct, but constant irradiance levels for each

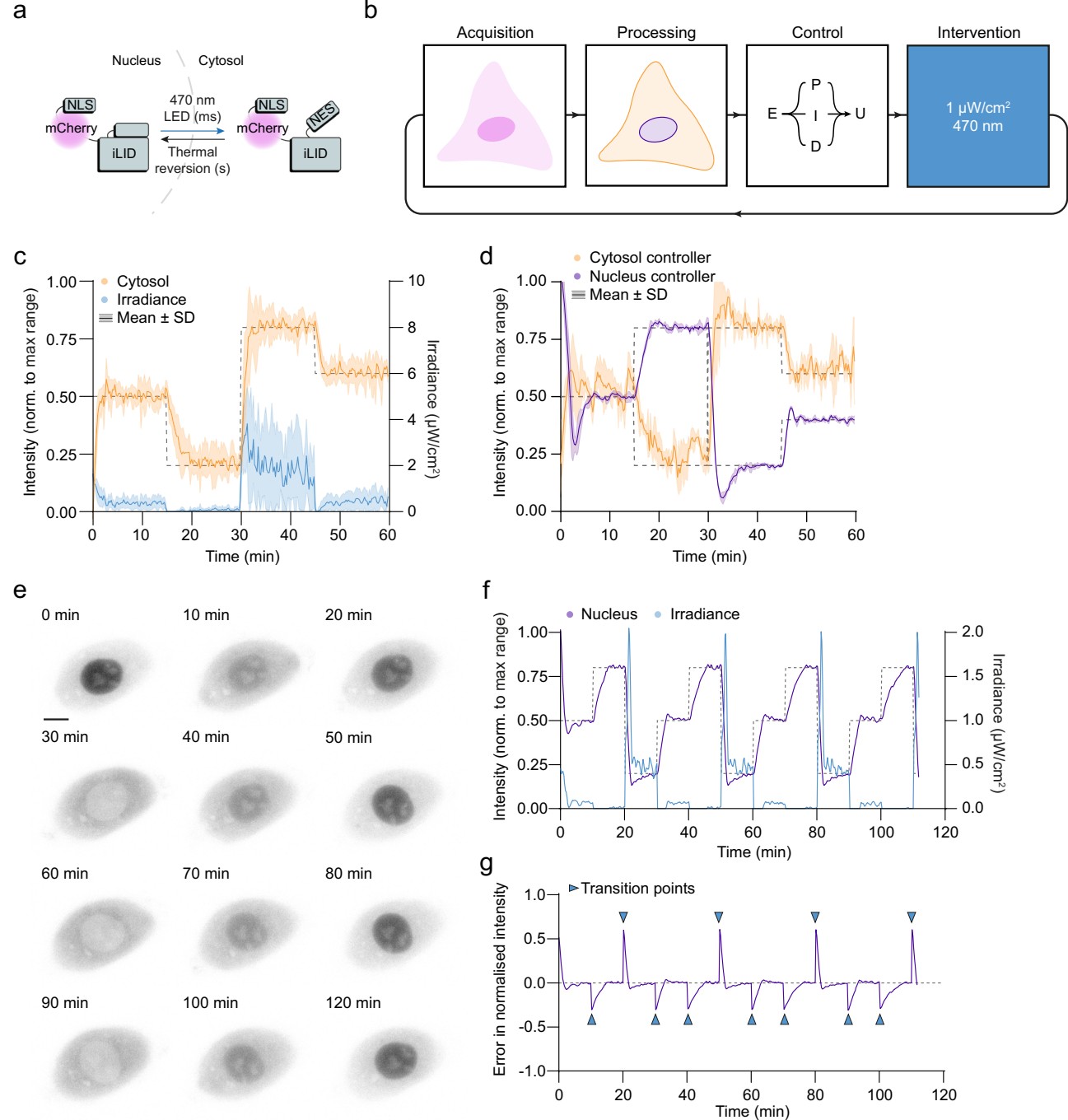

**Fig. 3 | Outcome-driven microscopy enables well-controlled titration of protein levels in the nucleus or cytosol. a** Schematic of the LEXY construct[77] used for optogenetic uncaging of a nuclear export sequence. **b** An outcome-driven microscopy approach to control of nucleocytoplasmic transport using a proportional-integral-derivative (PID) controller. **c** U2OS-LEXY cells were controlled using outcome-driven microscopy with a gain-scheduled PID controller to bring cytosolic intensities (orange) to specific setpoints (dashed line) by modulating irradiance over time (blue). Intensities are represented normalised to the maximum dynamic range. $n = 10$ cells. Error bands show mean ± standard deviation (SD). **d** A U2OS-LEXY cell controlled using controllers for cytosol intensity (orange) and nucleus intensity (purple) to various setpoints (dashed lines). $n = 3$ runs for each controller. Error bands show mean ± standard deviation (SD). **e** A U2OS-LEXY cell (grey) controlled using a nucleus intensity controller to multiple specific setpoints. Scale bar: 10 μm. **f** Normalised nucleus intensity and irradiance for the cell in (**e**), controlled to multiple setpoints over two hours. **g** The error in normalised intensity (intensity–setpoint) for the same cell as in (**e**). Transition points (changes of setpoint) are shown with blue arrowheads. Source data are provided as a Source Data file.

individual cell over time to maintain cells at the desired setpoint. This indicates that future approaches of outcome-driven microscopy could first find these stable values and then require a lower temporal resolution for continued long-term control. To verify that the fluorescence intensity in live-cell imaging is a suitable reporter for the intracellular protein concentration, we controlled multiple cells to different setpoints, fixed the cells on-stage, and visualised the shuttled protein using immunofluorescence (Supplementary Fig. 6c–e). The quantification of immunofluorescence corroborates that the fluorescence intensity measured during live-cell microscopy is a good reporter for

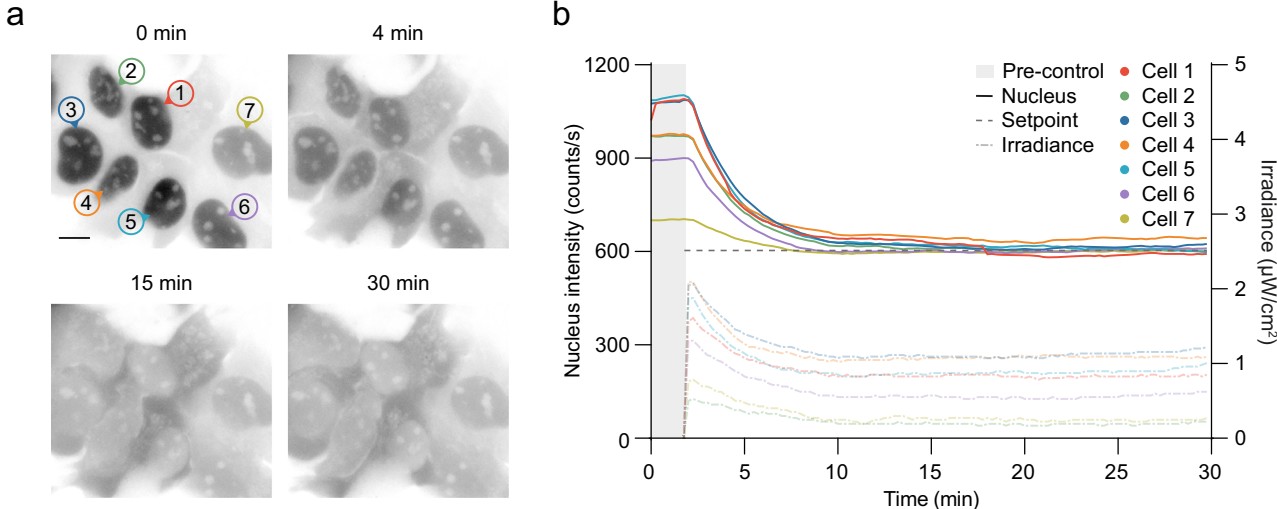

**Fig. 4 | Simultaneous intensity-based outcome-driven control of nucleocytoplasmic transport in multiple cells. a** Seven USOS-LEXY cells (grey) with various expression levels controlled simultaneously using pulse-width modulation to independently bring nucleus intensity to the same setpoint. **b** Nucleus intensities over time for the cells in (**a**), brought to the same setpoint (dashed grey line). Note that different intensities are required to bring each cell to the same intensity setpoint. Source data are provided as a Source Data file.

protein concentration. Together, these experiments pave the way for the controlled titration of protein levels in nucleus or cytosol, which could for example, be used to explore the dose-response functions of different transcription factors.

## Towards outcome-driven microscopy for active control of cell biology

In summary, we have established a proof-of-concept for the robust integration of smart microscopy and optogenetics to overcome cell-cell variation, bring cells to predefined outcomes, and achieve precise, outcome-driven control of microscopy experiments. This approach is also easily applicable to other existing microscope communication platforms that are used for smart microscopy[10,80–82], and the modularity allows for a diverse range of experiments, outcomes, and cell types.

While this platform establishes a flexible framework for outcome-driven cell control, practical considerations remain. First, when individually segmenting, analysing and controlling cells, increasing the number of cells controlled in parallel imposes greater computational demands. The computing time required for a full feedback loop scales linearly with the number of cells controlled (Supplementary Note 1). Second, effective outcome-driven control relies on the use of reversible optogenetic modules that must be carefully optimised and characterised for effective controller design. Third, hardware constraints such as the choice of spatial light modulator impacts spatial precision in both 2D and 3D applications, and may limit the ability to modulate light intensity independently and simultaneously across multiple regions and in thicker/heterogeneous samples, crucial for intensity-dependant processes in complex samples.

Advancing the field of smart microscopy by integrating active control of cell biology through optogenetics, we provide a platform for reproducible, cell-cell variation-independent, interrogation of biological processes. This outcome-driven approach can be further expanded into multi-process control and/or 3D environments, in order to provide insight into complex biological interactions and control morphodynamical transitions.

## Methods
### Cell culture
HT1080 (ATCC, CCL-121), MCF7 (ATCC, HTB-22), U2OS Flp-in T-Rex (a gift from Alessandro Sartori) and COS-7 (ATCC, CRL-1651) cells were cultured in DMEM high glucose medium (Capricorn, DMEM-HPSTA) supplemented with 9% foetal bovine serum (Corning, 35_079_CV) and 1% penicillin/streptomycin (GIBCO, 15140-122) at 37 °C with 5% $CO_2$. All cell lines were routinely screened (every 8–12 weeks) to ensure they were free from mycoplasma contamination.

The HT1080-TIAM cell line was produced for this study using lentiviral transduction. Lentivirus was produced by transfecting 10 cm dishes of HEK293T cells with 15 µg of the transfer plasmid pLenti_TIAM-tagRFP-SSPB-P2A-mVenus-iLID-CAAX (a gift from Mathieu Coppey), together with 10 µg psPAX2 lentivirus packaging plasmid and 5 µg lentivirus envelope plasmid (gifts from Didier Trono, Addgene #12260 and #12259, respectively) using 90 µL 1 mg/mL Polyethylenimine Hydrochloride (Polysciences, 24765). Growth media was harvested 24- and 48-h post-transfection, filtered with a 0.45 µm syringe filter (Corning, 431220) and virus was precipitated in precipitation solution (5× stock solution: 66.6 mM PEG 6000 (Thermo Scientific Chemicals, A17541.0B), 410 mM NaCl (Thermo Scientific Chemicals, 194090010), in ddH2O, pH 7.2). The viral supernatant was then centrifuged for 30 min at 1500 × *g* at 4 °C, and the pellet resuspended in 1× DPBS (Sigma-Aldrich, D8537) for storage at −80 °C. To generate HT1080-TIAM stable line, wild-type cells were seeded to a 24-well plate. Twenty-four hours later, the medium was refreshed with DMEM supplemented with 5 µg/mL polybrene (Sigma-Aldrich, TR-1003), and 5 µL viral suspension was added. The medium was refreshed after 24 h, and cells were first selected in complete medium with 20 µg/mL blasticidin (InvivoGen, 29-11-BL), before sorting for double-positivity with Fluorescence-Activated Cell Sorting (FACS) using a FACSAria™ Fusion Flow Cytometer (BD Biosciences) (Supplementary Fig. 7).

The U2OS-LEXY stable line was generated using flp/frt recombinase-mediated cassette exchange. Maternal U2OS Flp-in T-Rex were transfected with the donor plasmid pCDNA5-NLS-mCherry-LEXY and pOG44 (Invitrogen), carrying Flp recombinase. Isogenic cells stably expressing LEXY from a doxycycline-sensitive promoter were selected by treatment with 200 µg/ml hygromycin B (InvivoGen, ant-hg-5).

For cell migration experiments, HT1080-TIAM cells were seeded to 25 mm coverslips (VWR, 631-0172) coated with 200 µL 25 µg/mL fibronectin (Sigma-Aldrich, F1141) at low density, 6–18 h prior to imaging. 2 mM thymidine (Calbiochem, 6060) was added at the point of seeding to prevent division, and thymidine was kept in the medium for

imaging. For nucleocytoplasmic transport experiments, U2OS-LEXY cells were seeded to 25 mm coverslips at medium density, 24 h prior to imaging. 2 μg/mL doxycycline-hyclate (Abcam, ab141091) was added at the point of seeding to induce expression, and doxycycline was kept in the medium for imaging, unless otherwise described. Coverslips were mounted in Attofluor cell chambers (Invitrogen, A7816) for imaging.

For transient transfections when controlling other cell types, cells were seeded to 25 mm coverslips as described above. Twenty-four hours after seeding, MCF7 cells were transfected with 1 μg CMV_TIAM-tagRFP-SSPB-P2A-mVenus-iLID-CAAX (a gift from Mathieu Coppey) and COS-7 cells were transfected with 0.5 μg NLS-mCherry-LEXY (pDN122, a gift from Barbara Di Ventura & Roland Eils, Addgene #72655). FuGENE 6 transfection reagent (Promega, E2691) was used according to manufacturer's instructions, with 3 μL transfection reagent per 1 μg plasmid DNA.

For phototoxicity experiments, 0.25 μM SYTOX Deep Red Nucleic Acid Stain (Invitrogen, S11380) was added to cells 15 min before imaging. Two stage positions were imaged, one with constant LED illumination and one kept in the dark as a control.

### Cloning
The donor vector pCDNA5-NLS-mCherry-LEXY, a mammalian expression vector for LEXY, was derived from NLS-mCherry-LEXY, and pCDNA5/FRT/TO (Invitrogen) by conventional molecular cloning.

### Fixation and immunostaining
For quantification of protein localisation after outcome-driven control, cells were fixed on-stage with pre-warmed 4% paraformaldehyde (Thermo Scientific, 28906) for 10 min while maintaining LED illumination. Next, cells were washed with 1× DPBS and permeabilized with 0.2% Triton ×-100 (Sigma-Aldrich, ×100) for 10 min. Following further washing, blocking was performed with 3% bovine serum albumin (Carl Roth, 8076) for 30 min. Cells were incubated with primary antibody (rabbit anti-RFP; 1:500; Rockland, 600-401-379) for 2 h at room temperature, followed by washing and secondary antibody incubation (goat anti-rabbit Alexa Fluor 488; 1:1000; Thermo Fisher Scientific, A-11034) for 1 h at room temperature. Buffers were prepared in 1× DPBS, while antibodies were diluted in blocking buffer.

### Imaging hardware
Imaging was performed using a Nikon Ti dual-turret inverted microscope equipped with a 40× CFI Plan Fluor NA 1.3 oil immersion objective (Nikon, MRH01401), a sample incubator (Tokai-Hit), and a pco.edge cooled sCMOS camera (Excelitas).

For epifluorescence illumination, a pE-4000 LED illumination system (CoolLED) was used as a light source, and ET 514 nm Laser Bandpass (Chroma, 49905) and ET mCherry (Chroma, 49008) filter cubes were placed in the lower filter turret.

For optogenetic illumination, a 470 nm LED (Mightex BLS-LCS-0470-15-22) and a Polygon 400 digital mirror device (Mightex) were equipped for the light source, and a 10/90 beamsplitter (Chroma, 21012) was placed in the upper filter turret. Neutral density filters of optical density 0.5 and 0.9 (ThorLabs, NE205B and NE209B) were also placed into the light path to reduce LED intensity.

For multi-cell control of nucleocytoplasmic transport, a Nikon Ti2-E dual-turret inverted microscope was used, equipped with the above objective, sample incubator, and epifluorescence illumination and filter sets. A Sona-6 Extreme sCMOS camera (Andor, 4BV6U) was equipped. For optogenetic illumination, a pE-800 LED illumination system (CoolLED) was used with a Mosaic 3 digital mirror device (Andor). A neutral density filter of optical density 2.0 (ThorLabs, NE220B) was placed in the light path, and a 10/90 beamsplitter (ThorLabs, BSN10R) was placed in the upper filter turret.

A photodiode power sensor (ThorLabs, S121C sensor with a PM400 console) with built-in irradiance calibration was used to calculate irradiance at the sample plane. For cell migration experiments, images were acquired with a 60 s interval. For nucleocytoplasmic transport experiments, a 15 s interval was used.

### Smart microscopy platform for outcome-driven microscopy
Custom Python code was written for control of microscope hardware (Supplementary Note 1) in Python version 3.9.13. We built a modular platform, where an outcome-driven strategy can be defined using separate modules for software-hardware bridging, on-the-fly image analysis, and closed-loop biological control.

In this study, our microscope bridge module uses μ-Manager version 2.0[83] Nightly Builds 20220930 (Nikon Ti) & 20250310 (Nikon Ti-2E) to control the microscopes and peripheral components, and the Pycro-Manager version 0.19.2 package as a translation layer[84]. Our on-the-fly analysis module uses Segment Anything[75] with custom extensions (Supplementary Note 2), and our closed-loop control module uses either a direct orientation-correction controller (for cell migration), or a PID controller (for nucleocytoplasmic transport) (Supplementary Note 3).

### Post-experiment image and data processing
All post-experiment imaging processing was performed using FIJI 1.54f[85]. Data processing was performed using Python 3.9.13, GraphPad Prism 9.5.0, or Microsoft Excel. GraphPad Prism 9.5.0 and Adobe Illustrator 28.7 were used for data visualisation. MATLAB v.2023b was used for controller design.

Path deviation was determined by calculating the distance between the centroid of the cell and the nearest point on the path (the projected position) at each timepoint. Projected speed was determined by then dividing the distance between projected points by the imaging interval.

Expression levels of cells were calculated as the corrected total cell fluorescence: cell integrated density − (cell area * background mean grey value). Rates of cytosolic intensity change during export or import of LEXY were calculated by measuring cytosolic intensity at 15 s intervals with LED constantly on (10.4 μW/cm$^2$) or off, then fitting to a non-linear one phase decay in GraphPad Prism 9.5.0.

For phototoxicity experiments, SYTOX Deep Red positive cells were counted manually. The total number of cells was estimated by thresholding the image to segment all cells, measuring the total pixel area occupied by cell objects, and dividing this value by the average of 10 cell areas (in pixels).

For comparison between the standard and gain-scheduled PID controllers, mean error at steady-state was calculated by averaging the error from the point where the setpoint was reached. This point is estimated when the derivative of the signal reduced to zero or changed sign. In this way, we clean the error from the transition time.

Nucleus:cytosol intensity ratio, for quantifying of protein localisation after outcome-driven control, was calculated by dividing the (background corrected) mean grey value of the nucleus by that of the cytosol.

### Statistics and reproducibility
All information on sample sizes, number of replicates, and statistical analyses is provided in the corresponding figure legends. No statistical method was used to predetermine sample size. No data were excluded from the analyses. The experiments were not randomised, and investigators were not blinded to allocation during experiments and outcome assessment, as this was not necessary for the study.

### Reporting summary
Further information on research design is available in the Nature Portfolio Reporting Summary linked to this article.

## Data availability

All datasets generated and analysed during this study, including raw and processed image files (TIFF), quantitative data (CSV, XLSX), and custom analysis code (Python scripts), are available in the Yoda database[86]: https://doi.org/10.24416/UU01-M0U1OT. Source data are provided with this paper.

## Code availability

Code for our smart microscopy platform, including a guide for integrating outcome-driven microscopy, is available on GitHub: https://github.com/UU-cellbiology/UU_SmartMicroscopy. v.1.0.0 was used for the present study[87]: https://doi.org/10.5281/zenodo.14420261.

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

## Acknowledgements

We thank Eugene Katrukha (Utrecht University) and Dylan Kalisvaart (Delft University of Technology) for helpful discussions, Toni van Capel (Utrecht University) for help with cell sorting, Alessandro Sartori (University of Zurich) for the U2OS Flp-In T-Rex cell line, and Mathieu Coppey, Barbara Di Ventura, Roland Eils, and Didier Trono for sharing plasmids. This work was supported by the Netherlands Organization for Scientific Research (NWO) Gravitation programme IMAGINE! (Project

number 24.005.009) (to L.C.K., C.S.S.), by the National Roadmap Initiative NL-BioImaging-AM (to L.C.K.), and by the Eindhoven, Wageningen, Utrecht Alliance through the Centre for Living Technologies (to L.C.K.).

## Author contributions

J.B.P., W.N., F.B., and L.C.K. conceived the study. J.B.P., A.R., J.S., H.G.H., A.J.M.G., W.S., W.N. designed and performed experiments. A.R., J.S., M.T.P.L., V.J.W.H., and I.S. developed software. J.B.P., A.R., M.T.P.L., V.J.W.H., and I.S. analysed data. J.B.P., A.R., M.T.P.L., and V.J.W.H. prepared figures. J.B.P., A.R., and L.C.K. wrote the manuscript. C.S.S., L.C.K. contributed funding and supervised the study.

## Competing interests

The authors declare no competing interests.
