## [Peer Review File · Nature Communications]

Closed-loop optogenetic control of cell biology enables outcome-driven microscopy

Corresponding Author: Professor Lukas Kapitein

Version 0:

Reviewer comments:

Reviewer #1

(Remarks to the Author)

In the paper, the authors describe and implement a closed loop microscopy system to steer biological systems via optogenetics in a controlled way depending on a desired spatio temporal pattern. Specifically, the authors describe a modular software framework that is integrated into a microscope setup that allows to detect cells and apply spatially precise optogenetic stimulation to induce cell migration in a controlled way. This is achieved by segmenting the cell outline on the fly via a general segmentation model (Segment anything/SAM) and then to selectively illuminate (via a DMD) the cell border that is closest to a predefined setpoint trajectory thus guiding the cell motion along it. The power of this approach is demonstrated on steering multiple (three) cells simultaneously while incorporating live collision avoidance as an additional control objective. Finally, the approach is demonstrated on a different task of controlling the relative subcellular expression of a protein in the cytosolic and nuclear region of a cell.

Overall, the paper is well written and the experimental results are quite impressive in that they on original application of the powerful idea of closed loop steering of microscopes with on the fly-analysis outside of purely optimizing imaging conditions. The steering of multiple cells in particular is very impressive. As such, I think the paper is a very solid practical demonstration of a novel application of the concept of smart microscopy that should be of high interest to the community.

I only have two issues that I think should be addressed in the paper:

1) The authors write that they provide a "modular smart microscopy platform" but I think that the current state of the software/repository is more of a demonstration that really a framework that could be easily used by other researchers. Specifically,

- the modularization of the code happens in a single file (main.py), which makes it hard to see how different parts of the system relate to each other and could be reused. I would suggest to convert the repository into a more modular library that can be directly installed, rather than a single file
- some dependencies are missing (eg SAM is imported but not declares as dependency)

2) The paper could be improved by discussing the limitations of the approach or their experimental demonstration. Specifically,

- what is the limitation regarding number of cells that can be steered?
- what is the spatial resolution of the illumination device (eg DMD) required to successfully steer cells?

minor comments:

- data figshare link (<https://doi.org/10.6084/m9.figshare.26509702>) is broken
- ref SAM [74] arxiv ref should maybe be updated to the published version in ICCV 2023

(Remarks on code availability)

I had a look at the code organization

Reviewer #2

(Remarks to the Author)

In this manuscript, the authors present outcome-driven microscopy that combines AI-driven image analysis with optogenetic control in a closed-loop configuration. The study demonstrates that cell migration can be continuously guided along a predetermined trajectory without manual intervention and that protein expression levels can be modulated with subcellular precision. As stated in the manuscript, closed-loop control combined with optogenetics has been previously explored in both in vivo and in vitro by several groups, the contribution of this work lies in its automatic, subcellular control over dynamic cellular processes. However, additional experimental evidence is needed to demonstrate precise spatiotemporal control over cellular behavior.

One of the major challenges in studying cellular dynamics is the inherent stochasticity of protein expression and the heterogeneity within cell populations. The implementation of an automatic control system, as described in this study, offers the advantage of capturing a large number of events, which in turn enables a more robust statistical analysis of subpopulation behaviors and dynamic responses. The authors have successfully demonstrated multi-cell manipulation. It is reasonable to expect that, by scaling up the system, the number of cells that can be simultaneously monitored and controlled could increase by at least an order of magnitude compared to conventional techniques. Furthermore, by expanding the field of view, the system could potentially accommodate even larger cell populations, which would be a significant step forward in high-throughput dynamic cellular studies.

One aspect that requires further clarification is the methodology for optogenetic activation at the subcellular level. The manuscript does not provide detailed information regarding the illumination strategy to target specific cell regions. For instance, it remains unclear whether the area of illumination within the cytosol and nucleus is confined to a specific subregion or if it covers the entire compartment. Given that cellular kinetics may vary depending on the spatial distribution of the light stimulus and considering that cells are inherently three-dimensional structures where a two-dimensional illumination pattern might simultaneously affect both cytosolic and nuclear compartments—this ambiguity could have significant implications for data interpretation.

Another point of concern is the reliance on measured intensity values to regulate the system under the assumption that the expression level of the optogenetic construct remains constant. In long-term experiments, phototoxicity is a potential issue that could diminish cellular responsiveness, thereby introducing errors in dynamic measurements. The current demonstration involves only two cycles of intensity control. To better assess the robustness and reliability of the system, it would be advantageous to conduct extended, long-term control experiments that evaluate how phototoxic effects and potential changes in construct expression influence the system's performance over time.

The performance of the AI model used for real-time analysis is heavily dependent on the training dataset, and in this study, the authors have only validated their approach using two different cell types. This raises the question of whether the approach can be applied to other cellular systems. Expanding the experimental validation to include additional cell types would be important for establishing the broader applicability of the system.

While the authors demonstrate controlled titration of protein levels in the nucleus and cytosol based solely on intensity measurements, it would strengthen their conclusions if these measurements could be corroborated by alternative validation techniques. Furthermore, data presented in Figures 1h and 1i suggest that neither the speed of cell migration nor the changes in speed appear to be significantly dependent on the irradiance. A more detailed discussion is warranted to explain these observations, clarifying whether there are underlying mechanisms or compensatory processes at play that might account for the apparent independence from irradiance.

(Remarks on code availability)

Version 1:

Reviewer comments:

Reviewer #1

(Remarks to the Author)

In the revised version the authors have addressed all my original issues and I have no further concerns.

(Remarks on code availability)

Reviewer #2

(Remarks to the Author)

In the revised manuscript, the authors demonstrate simultaneous closed-loop control of seven U2OS-LEXY nuclei, where live cells are guided toward user-defined states in real-time. Such a system may have important applications in the study of cell biology. Nevertheless, we still have several concerns about this manuscript.

1. The authors have employed a nucleus-only illumination strategy to avoid cytosolic activation, but they do not quantify axial confinement. A confocal z-stack of a fluorescent calibration slide illuminated through the same optical path—or an optical simulation—would reveal the spread of light above and below the focal plane. Such data are particularly important if the technique is to be extended to thicker or more heterogeneous specimens.

2. Phototoxicity experiments are not so convincing. U2OS-LEXY cells were monitored for morphological changes during six hours of repeated activation, but viability was not assessed beyond cell shape, and no assay was performed on the HT1080 cells used in migration studies. Different cell types can show different sensitivity to blue light. We encourage the authors to include a standard ROS or live/dead stain for HT1080s and to extend the U2OS assay to twenty-four hours to mimic multi-day experiments.

3. Scalability is another concern in the revised manuscript. Showing control of seven nuclei is a useful proof of concept; however, readers still have no sense of the practical ceiling. They should provide some key parameters, such as frame-to-actuation latency and maximum sustainable frame rate as a function of cell count.

4. Another concern is the absence of an alternative validation that mCherry fluorescence intensity truly reports intracellular protein concentration over the time scales explored. All quantitative claims rely on this linear mapping. The authors argue that photobleaching correction is enough, yet they provide no biochemical evidence. We suggest that the authors fix a subset of cells at the end of a closed-loop experiment and assay protein levels by Western blot or quantitative immunofluorescence.

(Remarks on code availability)

Version 2:

Reviewer comments:

Reviewer #2

(Remarks to the Author)

The authors answered most of our questions. We only have one minor question. The revised manuscript includes a z-stack of the activation volume (Extended Data Fig. 6a), which is presented qualitatively. No quantitative axial spread metric, such as FWHM of the activation field, is reported. A brief quantification would strengthen claims about extensibility to thicker/heterogeneous samples.

(Remarks on code availability)

Version 3:

Reviewer comments:

Reviewer #2

(Remarks to the Author)

The authors addressed all our questions. We have no more questions.

(Remarks on code availability)

Outcome-Driven Microscopy: Closed-Loop Optogenetic Control of Cell Biology

Josiah B. Passmore, Alfredo Rates et al.

Point-by-point response to the reviewer comments

Reviewer #1 (Remarks to the Author):

In the paper, the authors describe and implement a closed loop microscopy system to steer biological systems via optogenetics in a controlled way depending on a desired spatio temporal pattern. Specifically, the authors describe a modular software framework that is integrated into a microscope setup that allows to detect cells and apply spatially precise optogenetic stimulation to induce cell migration in a controlled way. This is achieved by segmenting the cell outline on the fly via a general segmentation model (Segment anything/SAM) and then to selectively illuminate (via a DMD) the cell border that is closest to a predefined setpoint trajectory thus guiding the cell motion along it. The power of this approach is demonstrated on steering multiple (three) cells simultaneously while incorporating live collision avoidance as an additional control objective. Finally, the approach is demonstrated on a different task of controlling the relative subcellular expression of a protein in the cytosolic and nuclear region of a cell.

Overall, the paper is well written and the experimental results are quite impressive in that they on original application of the powerful idea of closed loop steering of microscopes with on the fly-analysis outside of purely optimizing imaging conditions. The steering of multiple cells in particular is very impressive. As such, I think the paper is a very solid practical demonstration of a novel application of the concept of smart microscopy that should be of high interest to the community. I only have two issues that I think should be addressed in the paper:

1. *The authors write that they provide a "modular smart microscopy platform" but I think that the current state of the software/repository is more of a demonstration that really a framework that could be easily used by other researchers. Specifically,*

- the modularization of the code happens in a single file (main.py), which makes it hard to see how different parts of the system relate to each other and could be reused. I would suggest to convert the repository into a more modular library that can be directly installed, rather than a single file

- some dependencies are missing (eg SAM is imported but not declares as dependency)

- We thank the reviewer for their positive assessment and constructive comments. Following the reviewer's suggestion, we have changed the structure of our software implementation to follow a package structure, with a setup.py installation file that includes all the required libraries. We also renamed the previous main.py as an use-case. It is important to note that the main file just coordinates the modules, while each module can still be imported and used separately. To further demonstrate this, we have added an example simplified implementation to the github repository.
- In addition, we have improved our explanations of dependencies and package structure in the readme. Some packages such as SAM, although contained now in the setup.py file, are only a dependency for specific modules depending on the implementation.

2. *The paper could be improved by discussing the limitations of the approach or their experimental demonstration. Specifically,*

- *what is the limitation regarding number of cells that can be steered?*
- *what is the spatial resolution of the illumination device (eg DMD) required to successfully steer cells?*

- As recommended by the reviewer, we have added a section to the text to discuss the limitations and practical considerations of our approach (See lines 263-271).
- To address the question about the number of cells that can be controlled, we have performed additional experiments aimed at controlling more cells and exploring limitations of control. This was possible by implementing a new microscope system equipped with a DMD module that could cover a larger field of view, but also independently modulate light intensity for individual cells using pulse-width modulation. Using this implementation, we have now also developed multicellular control for the experiments to control nuclear/cytoplasmic intensities, and demonstrate independent intensity-driven control of seven cells (See Figure 4 and lines 236-247).
- With respect to the spatial resolution required to successfully the steer cells, earlier work has already shown that cell guidance can also be achieved with a single spot of focused light (e.g. [10.1038/nature08241](https://doi.org/10.1038/nature08241), [10.1371/journal.pbio.3002307](https://doi.org/10.1371/journal.pbio.3002307)). This indicates that the spatial resolution of the illumination device is not a factor for these experiments, but we do envisage experiments where this could be important (e.g. precise control of individual cells in crowded environments or in the third dimension) as such, we have included this in our discussion of practical considerations.

minor comments:

- data figshare link (<https://doi.org/10.6084/m9.figshare.26509702>) is broken

- We thank the reviewer for bringing this to our attention. We have since fixed the above link (for our preprinted version), but have replaced the link with another for the full dataset including experiments for this revision (<https://doi.org/10.24416/UU01-MOU1OT>).

- ref SAM [74] arxiv ref should maybe be updated to the published version in ICCV 2023

- We have updated the reference to SAM as suggested.

Reviewer #2 (Remarks to the Author):

In this manuscript, the authors present outcome-driven microscopy that combines AI-driven image analysis with optogenetic control in a closed-loop configuration. The study demonstrates that cell migration can be continuously guided along a predetermined trajectory without manual intervention and that protein expression levels can be modulated with subcellular precision. As stated in the manuscript, closed-loop control combined with optogenetics has been previously explored in both in vivo and in vitro by several groups, the contribution of this work lies in its automatic, subcellular control over dynamic cellular processes. However, additional experimental evidence is needed to demonstrate precise spatiotemporal control over cellular behavior.

1. *One of the major challenges in studying cellular dynamics is the inherent stochasticity of protein expression and the heterogeneity within cell populations. The implementation of an automatic control system, as described in this study, offers the advantage of capturing a large number of events, which in turn enables a more robust statistical analysis of subpopulation behaviors and dynamic responses. The authors have successfully demonstrated multi-cell manipulation. It is reasonable to expect that, by scaling up the system, the number of cells that can be simultaneously monitored and controlled could increase by at least an order of magnitude compared to conventional techniques. Furthermore, by expanding the field of view, the system could potentially accommodate even larger cell populations, which would be a significant step forward in high-throughput dynamic cellular studies.*
 - We thank the reviewer for the enthusiastic and constructive feedback on our manuscript. For our revised manuscript, we implemented an additional microscope system that enabled us to control larger number of cells simultaneously with constant monitoring and analysis (See Figure 4 and lines 236-247).
2. *One aspect that requires further clarification is the methodology for optogenetic activation at the subcellular level. The manuscript does not provide detailed information regarding the illumination strategy to target specific cell regions. For instance, it remains unclear whether the area of illumination within the cytosol and nucleus is confined to a specific subregion or if it covers the entire compartment. Given that cellular kinetics may vary depending on the spatial distribution of the light stimulus and considering that cells are inherently three-dimensional structures where a two-dimensional illumination pattern might simultaneously affect both cytosolic and nuclear compartments—this ambiguity could have significant implications for data interpretation.*
 - We thank the reviewer for raising this point. It is correct to assume that the area of illumination has an effect on the kinetics of response to light intensity. Illumination of the whole cell vs nucleus ensures that optogenetic modules are held in the activated state throughout the cell during control. As such, during the development of our method, we used global illumination to control protein levels, and have added an explanation of this to the main text (See lines 175-177).
 - However, for multi-cell control of nuclear intensity, we chose to illuminate the nuclei only, in order to prevent unintended illumination due to light scattering when controlling cells in close proximity. We have added an explanation of this to the text (See lines 240-242). In both cases, global illumination and nucleus illumination, the control strategy drives the concentration to the desired setpoint without significant differences.
 - Indeed, a two-dimensional light pattern illuminating just the nucleus is expected to also affect some of the cytosolic compartment. However, as we do not derive intrinsic biological parameters from this control, we do not consider this to significantly affect our conclusions.
3. *Another point of concern is the reliance on measured intensity values to regulate the system under the assumption that the expression level of the optogenetic construct remains constant. In long-term experiments, phototoxicity is a potential issue that could diminish cellular responsiveness, thereby introducing errors in dynamic measurements. The current demonstration involves only two cycles of intensity control. To better assess the robustness and reliability of the system, it would be advantageous to conduct extended, long-term control experiments that evaluate how phototoxic effects and potential changes in construct expression influence the system's performance over time.*

- As we normalize the measured intensity values with the dynamic range of the cell, we do not expect changes in fluorescence, either due to expression or photobleaching, to significantly affect our control. To address this point, we now included extended, long-term control experiments, both with an increased number of setpoints, and maintenance of a single setpoint long-term during expected expression increase and stability (See Fig. 3e-g, Extended data Fig. 5d-e, and lines 228-235).
 - We have furthermore examined phototoxicity due to blue-light exposure and found no phototoxic effects within the timescales and irradiance levels used in this study (See Extended Data Fig. 3d and lines 197-200)
4. *The performance of the AI model used for real-time analysis is heavily dependent on the training dataset, and in this study, the authors have only validated their approach using two different cell types. This raises the question of whether the approach can be applied to other cellular systems. Expanding the experimental validation to include additional cell types would be important for establishing the broader applicability of the system.*
- The AI model that we used for cell segmentation was a generic pretrained model and is not expected to differ by cell type. However, to validate our segmentation and control strategy on other cell types, we have included demonstration of control of an additional cell type for cell migration (See Supplementary Video 5 and lines 146-150), and an additional cell type for nucleocytoplasmic transport (See Extended Data Fig. 5e and lines 233-235).
5. *While the authors demonstrate controlled titration of protein levels in the nucleus and cytosol based solely on intensity measurements, it would strengthen their conclusions if these measurements could be corroborated by alternative validation techniques.*
- We were unsure how to experimentally address this comment. In our work, we demonstrate that we can dynamically control protein levels in specific cellular compartments by controlled exposure to blue light using an approach that is tailored to individual cells to account for variations in overall expression of the LEXY construct. It is not possible to directly measure protein concentration in these assays, as we control live single cells on the fly in a non-invasive way. Because these processes are controlled dynamically using a light microscope, intensity measurements are the only way to assess controller success.
 - Any alternative validation technique that we could conceive to validate protein levels without intensity-based measurements would involve biochemical assays performed after the experiment, but this poses some problems. First of all, when blue-light exposure is aborted the LEXY construct will quickly return to the nucleus, so the desired level is only maintained during the actual microscopy experiment. Second, even if we could rapidly fix cells during the experiment and perform subsequent nuclear purification and biochemical analysis, this would require bulk experiments controlling all cells in the sample, while we can only control a relatively small number of cells compared to the entire population.
6. Furthermore, data presented in Figures 1h and 1i suggest that neither the speed of cell migration nor the changes in speed appear to be significantly dependent on the irradiance. A more detailed discussion is warranted to explain these observations, clarifying whether there

are underlying mechanisms or compensatory processes at play that might account for the apparent independence from irradiance.

- We find the relationship between irradiance and cell speed very intriguing, and indeed expect additional underlying mechanisms to be in play. As suggested, we have added an additional discussion about this to the main text (See lines 126-132).

Outcome-Driven Microscopy: Closed-Loop Optogenetic Control of Cell Biology

Josiah B. Passmore, Alfredo Rates *et al.*

Point-by-point response to the reviewer comments (second round of revision)

Reviewer #1 (Remarks to the Author):

In the revised version the authors have addressed all my original issues and I have no further concerns.

- We thank the reviewer for their positive assessment and constructive comments.

Reviewer #2 (Remarks to the Author):

In the revised manuscript, the authors demonstrate simultaneous closed-loop control of seven U2OS-LEXY nuclei, where live cells are guided toward user-defined states in real-time. Such a system may have important applications in the study of cell biology. Nevertheless, we still have several concerns about this manuscript.

1. The authors have employed a nucleus-only illumination strategy to avoid cytosolic activation, but they do not quantify axial confinement. A confocal z-stack of a fluorescent calibration slide illuminated through the same optical path—or an optical simulation—would reveal the spread of light above and below the focal plane. Such data are particularly important if the technique is to be extended to thicker or more heterogeneous specimens.

- We would like to clarify that the goal of nucleus-only illumination was not to avoid cytosolic activation, but to limit accidental activation of neighboring cells. Axial confinement could indeed be important for more complex 3D samples, but is not very critical in our 2D experiments. Furthermore, all cells are illuminated in the same way, so the controller already takes into account effects like incomplete nucleus illumination or accidental cytosol illumination above/below the nucleus. We now include a confocal z-stack of the activation volume in Extended Data Figure 6A and have adjusted our wording in the main text for clarity (lines 249-250).

2. Phototoxicity experiments are not so convincing. U2OS-LEXY cells were monitored for morphological changes during six hours of repeated activation, but viability was not assessed beyond cell shape, and no assay was performed on the HT1080 cells used in migration studies. Different cell types can show different sensitivity to blue light. We encourage the authors to include a standard ROS or live/dead stain for HT1080s and to extend the U2OS assay to twenty-four hours to mimic multi-day experiments.

- We have replaced the 6-hour morphological analysis with more robust 24-hour viability assessments using live/dead staining. This was performed for both U2OS (Extended Data Figure 3d-f, lines 206-207) and HT1080 cells (Extended Data Figure 2h-j, lines 255-260). Our results show that optogenetic activation does not strongly affect the live/dead cell ratio over a 24-hour period.

3. Scalability is another concern in the revised manuscript. Showing control of seven nuclei is a useful proof of concept; however, readers still have no sense of the practical ceiling. They should provide

some key parameters, such as frame-to-actuation latency and maximum sustainable frame rate as a function of cell count.

- In the Supplementary Text (section 1.9), we now provide a discussion on how key experimental parameters scale with the number of cells to be controlled. We also clarify in the main text (lines 279-280) that the computing time scales linearly with the number of cells controlled.

4. Another concern is the absence of an alternative validation that mCherry fluorescence intensity truly reports intracellular protein concentration over the time scales explored. All quantitative claims rely on this linear mapping. The authors argue that photobleaching correction is enough, yet they provide no biochemical evidence. We suggest that the authors fix a subset of cells at the end of a closed-loop experiment and assay protein levels by Western blot or quantitative immunofluorescence.

- Because we only control a small subset of all cells presents on the coverslip, Western blotting of the lysates from all cells of the coverslip will not be able to assess protein level changes in the few cells that were controlled. To address the reviewer's concern, we have now performed quantitative immunofluorescence analysis on cells that were controlled to specific levels using optogenetics, as well as neighboring cells that were not controlled (Extended Data Figure 6b-d, lines 255-260). This revealed a strong similarity between the nuclear/cytosolic ratios obtained during live-cell imaging and those obtained from quantitative immunofluorescence.

Outcome-Driven Microscopy: Closed-Loop Optogenetic Control of Cell Biology

Josiah B. Passmore, Alfredo Rates *et al.*

Point-by-point response to the reviewer comments (third round of revision)

Reviewer #2 (Remarks to the Author):

The authors answered most of our questions. We only have one minor question. The revised manuscript includes a z-stack of the activation volume (Extended Data Fig. 6a), which is presented qualitatively. No quantitative axial spread metric, such as FWHM of the activation field, is reported. A brief quantification would strengthen claims about extensibility to thicker/heterogeneous samples.

- As requested by the reviewer, we have added a quantification (line profile) of the activation field in Extended Data Figure 6b.